# Effects of Pulsatile Flow Rate and Shunt Ratio in Bifurcated Distal Arteries on Hemodynamic Characteristics Involved in Two Patient-Specific Internal Carotid Artery Sidewall Aneurysms: A Numerical Study

**DOI:** 10.3390/bioengineering9070326

**Published:** 2022-07-18

**Authors:** Hang Yi, Mark Johnson, Luke C. Bramlage, Bryan Ludwig, Zifeng Yang

**Affiliations:** 1Department of Mechanical and Materials Engineering, Wright State University, Dayton, OH 45435, USA; mark.johnson@wright.edu; 2Division of NeuroInterventional Surgery, Department of Neurology, Wright State University/Premier Health—Clinical Neuroscience Institute, 30E. Apple St., Dayton, OH 45409, USA; lcbramlage@premierhealth.com (L.C.B.); brludwig@premierhealth.com (B.L.); 3Boonshoft School of Medicine, Wright State University, Dayton, OH 45435, USA

**Keywords:** internal carotid artery sidewall aneurysm (ICASA), hemodynamic behaviors, computational fluid dynamics (CFD), pulsatile flow rate (PFR), bifurcated shunt ratio, wall shear stress (*WSS*), oscillatory shear index (*OSI*), time-averaged pressure (*TAP*)

## Abstract

The pulsatile flow rate (PFR) in the cerebral artery system and shunt ratios in bifurcated arteries are two patient-specific parameters that may affect the hemodynamic characteristics in the pathobiology of cerebral aneurysms, which needs to be identified comprehensively. Accordingly, a systematic study was employed to study the effects of pulsatile flow rate (i.e., PFR−I, PFR−II, and PFR−III) and shunt ratio (i.e., 75:25 and 64:36) in bifurcated distal arteries, and transient cardiac pulsatile waveform on hemodynamic patterns in two internal carotid artery sidewall aneurysm models using computational fluid dynamics (CFD) modeling. Numerical results indicate that larger PFRs can cause higher wall shear stress (WSS) in some local regions of the aneurysmal dome that may increase the probability of small/secondary aneurysm generation than under smaller PFRs. The low WSS and relatively high oscillatory shear index (OSI) could appear under a smaller PFR, increasing the potential risk of aneurysmal sac growth and rupture. However, the variances in PFRs and bifurcated shunt ratios have rare impacts on the time-average pressure (TAP) distributions on the aneurysmal sac, although a higher PFR can contribute more to the pressure increase in the ICASA−1 dome due to the relatively stronger impingement by the redirected bloodstream than in ICASA−2. CFD simulations also show that the variances of shunt ratios in bifurcated distal arteries have rare impacts on the hemodynamic characteristics in the sacs, mainly because the bifurcated location is not close enough to the sac in present models. Furthermore, it has been found that the vortex location plays a major role in the temporal and spatial distribution of the WSS on the luminal wall, varying significantly with the cardiac period.

## 1. Introduction

Intracranial arterial walls have the probability of generating cerebral aneurysms (CAs) by aberrant focal dilatations [1,2,3,4], which may lead to unexpected consequences (e.g., stroke, coma, and/or death) if the aneurysmal dome ruptures [5,6,7,8]. Indeed, it has been estimated that 90% of spontaneous subarachnoid hemorrhages (SAHs) can be attributed to the rupture of cerebral aneurysms, which range in size from less than 5 mm to more than 25 mm patient-specifically [9,10]. This lack of clear guidance to evaluate unruptured CAs may lead to a false sense of security when withholding aneurysm treatment in an aneurysm misidentified as benign, or an unnecessary repair of more benign aneurysms which are misidentified as high risk for rupture. Not all patients with unruptured cerebral aneurysms are necessary to treat surgically, and treatment has been a matter of debate for many decades [10,11,12]. Thus, a comprehensive understanding of the pathophysiology of CAs is extremely important for physicians to evaluate treatments more responsibly. Specifically, hemodynamic characteristics, i.e., flow patterns, wall shear stress (WSS), oscillatory shear index (OSI), and time-averaged pressure (*TAP*), play an important role in the formation, growth, and rupture of cerebral aneurysms, and should be identified precisely using qualitative and quantitative manners [13,14,15,16,17]. So far, the risks associated with various factors (i.e., aneurysmal dome locations, high-risk aneurysmal morphologies, pre- and post-treatment states, and arterial blood flow conditions) on the pathophysiology of CAs have been estimated, to some extent, using different research strategies (i.e., in vivo, in vitro, and in silico), which were summarized by previous efforts [18,19,20,21,22,23,24,25]. Nevertheless, the pulsatile flow rates are typically patient-specific [26,27], and many studies had tentative investigations on how the varying pulsatile blood flow rates/pressures will influence hemodynamic characteristics (i.e., WSS and OSI) in cerebral arteries or CAs [28,29,30,31,32,33]. Sekhame and Mansour [28] studied time-averaged WSS distributions in the internal carotid artery (ICA) rather than in CA with three pulsatile flow rates using COMSOL Multiphysics (COMSOL Inc., Stockholm, Sweden) and found that waveform boundary conditions have important effects on the overall instantaneous hemodynamic factors assessed on the geometries, while the time-averaged WSS was constant for the studied cases. Sarrami-Foroushani et al. [29] studied the hemodynamic patterns in a ruptured ICA aneurysm with varying flow waveforms and discovered that systolic and time-averaged WSS and pressure on the aneurysm wall showed a proportional evolution with the mainstream flow rate. However, some of the above-mentioned studies reconstructed the models based on the image scanned from a ruptured aneurysm, which could be extremely different from the morphology of the realistic aneurysm. Furthermore, the mesh independence test in the simulations was not investigated, which may lead to significant differences in the results among different research groups despite the same/similar geometries, boundary conditions, and blood properties adopted. More importantly, another two parameters, i.e., the detailed flow field (i.e., velocity gradient and vortices) in the aneurysmal lumen and the variations of shunt ratios in bifurcated distal arteries with the simultaneously varied inlet flow rates have not been investigated in previous studies. Moreover, since the shapes and locations of CAs are always patient-specific, more investigations are still needed to enrich the intracranial aneurysm research community to better understand the pathophysiology of CAs.

Compared to the in vivo and in vitro experiments, the computational fluid dynamics (CFD) based in silico methods adopt an accessible and noninvasive manner to predict the blood flow patterns in CAs. Specifically, using reconstructed patient-specific artery and aneurysm models based on computed tomography (CT), X-rays, magnetic resonance (MRI), and digital subtraction angiography (DSA) scanned images as well as machine learning and deep learning algorithms [34,35,36,37], numerical modeling employs physiologically initial/boundary conditions, which can aid in identifying major translational knowledge gaps and provide a platform for implementing and evaluating potential solutions [14,20,38,39,40,41,42,43,44,45,46]. This approach offers several advantages, including the ability to: (1) study a system or phenomenon at different spatial and temporal scales in the aneurysmal sac, (2) perform analysis under varied conditions, i.e., model-based realistic blood flow conditions and arterial geometries, (3) evaluate critical situations in a noninvasive way, and (4) carry out cost-effective and high-fidelity studies, which can accelerate the better understanding of hemodynamic factors affecting the generation, growth, and rupture of the aneurysmal domes than in vivo/in vitro.

The objectives of this study are to employ CFD modeling to quantify the effects of pulsatile flow rate and shunt ratios in bifurcated arteries on the hemodynamic characteristics in two patient-specific internal carotid artery sidewall aneurysm (ICASA) models (e.g., ICASA−1 and ICASA−2). Specifically, a systematically parametric study was used to study the effects of pulsatile flow rate, shunt ratio in bifurcated distal arteries, and transient cardiac pulsatile flow on hemodynamic transport behaviors in ICASA−1 and ICASA−2 models. The novel results obtained in this study can contribute to filling the intracranial aneurysm community’s knowledge gaps about how blood flow patterns in the aneurysmal sac can be affected by different patient-specific pulsatile flow rates and shunt ratios in bifurcated distal arteries.

## 2. Numerical Methodology

### 2.1. Geometry and Mesh

To study the influences of the flow rate and shunt ratio on hemodynamic characteristics in ICASA models, two patient-specific cerebral aneurysm models (see Figure 1), i.e., (a) ICASA−1 (73-year-old, female), and (b) ICASA−2 (35-year-old, female), were built using an open source code, i.e., 3D Slicer, based on medical data provided by Miami Valley Hospital (Dayton, OH, USA) [47,48]. In the ICASA−1 model, the blood flows in through the ICA and flows out from the bifurcated distal arteries, i.e., posterior communicating artery (PComA) and ICA distal. In the ICASA−2 model, the blood flows in through the ICA to its distal bifurcation, i.e., middle cerebral artery (MCA) and anterior cerebral artery (ACA). In this study, the ICA distal and MCA are denoted by A1, and PComA and ACA are denoted by A2 (see Figure 1), respectively, for simplicity.

Two sets of poly-hexcore meshes have been generated for each aneurysm model using ANSYS Fluent Meshing 2021 R2 (Ansys Inc., Canonsburg, PA, USA), with different mesh sizes for the mesh independence test. Mesh details for the ICASA−1 and ICASA−2 models are shown in Table 1 and Figure 1. Regions (i.e., aneurysm sac) with potential rupture risks were discretized with refined mesh elements. Mesh independence tests were investigated by the comparisons of nondimensionalized velocity profiles $V_{i-i^{\prime }}^{*}$ at selected lines (see Figure 2), i.e., AA′ and BB′, in the two ICASA models, respectively, with a constant blood flow rate of 1.5 mL/s, density of 1.05 × 10^3^ kg/m^3^, and viscosity of 3.5 × 10^−3^ kg/m−s. Moreover, the WSS magnitudes on the edge of selected planes, i.e., plane A−A′ and plane B−B′, were compared in the generated meshes with the corresponded ICASA models to obtain the final mesh for parametric studies (see Figure 2). The equations for nondimensionalized velocity $V_{i-i^{\prime }}^{*}$ and nondimensionalized length, i.e., $\;L_{i-i^{\prime }}^{*}$ and $L_{i^{\prime \prime }-i^{\prime \prime }}^{*}$, can be expressed by:(1)$$V_{i-i^{\prime }}^{*}=\frac{\left|V\right|}{\left|V_{in}\right|}$$
(2)$$L_{i-i^{\prime }}^{*}=\frac{l^{\prime }}{L_{ii^{\prime }}}$$
(3)$$L_{i^{\prime \prime }-i^{\prime \prime }}^{*}=\frac{l^{\prime \prime }}{L_{i^{\prime \prime }i^{\prime \prime }}}$$
where $V_{in}$ is the inlet velocity, and V is the velocity for the selected lines $A\to A^{\prime }$ and $B\to B^{\prime }$ for the two ICASA models, respectively. l denotes the length of lines AA′ and BB′ from $A\to A^{\prime }$ and $B\to B^{\prime }$, separately. $l^{\prime \prime }$ is the arc length from point $A^{\prime \prime }\to$ $A^{\prime \prime }$ and $B^{\prime \prime }\to$ $B^{\prime \prime }$, respectively. In addition, $L_{i^{\prime \prime }i^{\prime \prime }}$ represents the perimeter of the arcs $A^{\prime \prime }A^{\prime \prime }$ and $B^{\prime \prime }B^{\prime \prime }$, separately.

Using generated meshes (see Table 1), the nondimensionalized velocities and WSS are shown in Figure 2. It can be observed that mesh 01 and mesh 04 are too coarse to generate accurate results. The variations in simulated velocities and WSS are within 1% between mesh 02 and mesh 03. Similar results can also be found with the comparisons in mesh 05 and mesh 06. Thus, based on the optimal balance between computational efficiency and accuracy, mesh 02 (i.e., with 2,741,603 elements, 25 prism layers, 3 peel layers, and size growth rate 1.05) and mesh 05 (i.e., with 3,012,970 elements, 20 prism layers, 3 peel layers, and size growth rate 1.05) were selected as the final meshes for each ICASA model to study the hemodynamic behaviors.

### 2.2. Governing Equations

The patient-specific pulsatile blood flow is always unsteady under the action of periodic pulsatile flow conditions. The continuity and momentum equations can be written in tensor form, i.e.,
(4)$$\frac{\partial u_{i}}{\partial x_{i}}=0$$
(5)$$\rho \frac{\partial u_{i}}{\partial t}+\rho \frac{\partial \left(u_{j}u_{i}\right)}{\partial x_{j}}=-\frac{\partial p}{\partial x_{i}}+\mu \frac{\partial }{\partial x_{j}}\left[\left(\frac{\partial u_{i}}{\partial x_{j}}+\frac{\partial u_{j}}{\partial x_{i}}\right)\right]+\rho g_{i}$$
where $u_{j}$ represents the blood flow velocity, *p* is the pressure, $g_{j}$ is the gravity, μ is blood dynamic viscosity. In this study, the flow regime is assumed as incompressible and Newtonian. The blood density of 1.05 × 10^3^ kg/m^3^ and dynamic viscosity of 3.5 × 10^−3^ kg/m−s were used. The blood flow in the cerebral and artery and aneurysm system has been identified as laminar, which was employed in this study (Reynolds number < 800). It is worth mentioning that the employed CFD method to simulate blood laminar flow in the cerebral arteries has been well validated with benchmarked experimental data in previous publications [49,50,51,52,53,54,55] by the good agreements in comparisons of flow field quantifications.

### 2.3. Wall Shear Stress (WSS)

Wall shear stress (WSS) plays a significant role in the formation, growth, and rupture of cerebral aneurysms, which is a tangential frictional force on the arterial wall and induced by the blood shearing flow. The equation to compute WSS can be written as, i.e.,
(6)$$WSS=\mu {\left(\frac{\partial u}{\partial y}\right)}_{y=0}$$
where u is the blood velocity parallel to the arterial wall, and *y* is the normal distance to the arterial wall. To more specifically analyze the WSS effects on the aneurysm sac wall (i.e., S1 in ICASA−1 and S2 in ICASA−2, shown in Figure 1), WSS was divided into three components representing the WSS vector with respect to Cartesian coordinates as follows, e.g.,
(7)$$WSS=WSS_{x}\vec{l}+WSS_{y}\vec{j}+WSS_{z}\vec{k}$$
where $WSS_{x}$, $WSS_{y}$, $WSS_{z}$ represent wall stress components in the *X*, *Y*, and *Z* directions in Cartesian coordinates. In this study, the surface-average wall shear stress $\bar{WSS}$ in the local aneurysmal region (e.g., R1 in ICASA−1 and R2 in ICASA−2, shown in Figure 1) was also employed to analyze the effects of pulsatile flow rates and shunt ratios in bifurcated distal arteries on intra-aneurysmal hemodynamic characteristics, i.e.,
(8)$$\bar{WSS}=\frac{1}{S}\oint \left|WSS\right|ds$$
where S is the surface area of the selected aneurysmal region, and s is the differential artery wall.

### 2.4. Oscillatory Shear Index (OSI)

Oscillatory shear index (OSI) describes the oscillating features during a pulsatile cycle that is characterized by a nondimensional parameter. It is often employed to describe the disturbance of the blood flow field in the aneurysm. OSI also shows the magnitude of WSS alterations and illustrates the oscillation of tangential force in one cardiac cycle [56]. The equation to calculate OSI is, e.g.,
(9)$$OSI=\frac{1}{2}\left(1-\frac{|\int_{0}^{T}WSSdt|}{\int_{0}^{T}\left|WSS\right|dt}\right)$$
where T is the period of one cardiac cycle, i.e., 1.0 s in this study (see Figure 3). The OSI varies between 0 and 0.5, with a value of 0 observed in regions of unidirectional flow and a value of 0.5 observed in regions of fully oscillatory flows.

### 2.5. Time-Averaged Pressure (TAP)

It has been recognized that high blood pressure can be the major risk factor for the pathophysiology of CAs [57] since the induced hemodynamic stress and inflammation by high blood pressure could lead to arterial wall damage and dilation, resulting in the growth and rupture of cerebral aneurysms [58,59]. In this study, the time-averaged pressure (TAP) was calculated under different PFRs and shunt ratios in bifurcated distal arteries, i.e., written as,
(10)$$TAP=\frac{\int_{0}^{T}pdt}{T}$$

### 2.6. Boundary and Initial Conditions

To systematically investigate the PFR effects on the intra-hemodynamic characteristics in the ICASA models, three transient pulsatile flow rate waveforms (i.e., PFR−I, PFR−II, and PFR−III) with a period of T = 1.0 s were applied as the boundary conditions at the ICA inlet (see Figure 3), representing cardiac PFR conditions. Specifically, the waveform of PFR−II is based on pulsatile flow rates obtained from a validated 1D model, which has been studied in our previous investigations [47,60]. According to a statistical study determining blood flow rates in cerebral arteries [26], another two waveforms, i.e., PFR−I and PFR−III, were generated by multiplying 0.57 and 1.43 times of PFR−II, respectively, which fall into the 95% confidence interval of blood flow rate in ICA based on statistics [26]. It has been found in a previous study that the flow rate variation in PComA is insignificant, and the bifurcated shunt ratio (i.e., flow rate ratio) for ICASA−1 in PComA and ICA distal is approximately 25:75 [61]. Thus, this study only investigated the effects of shunt ratios on hemodynamic patterns in the ICASA−2 model. Specifically, the volumetric shunt ratios (i.e., the minimum and the maximum), q_A1_ vs. q_A2_, e.g., 64:36 [26] and 75:25 [62], were used to investigate the effects of flow-splitting variances in bifurcated distal arteries on hemodynamic behaviors. Additionally, the arterial walls are assumed to be stationary and non-slip, and the blood circulation system is operated under the pressure waveform (see Figure 3) obtained by previous studies [47,60].

### 2.7. Numerical Settings

CFD simulations were executed using Ansys Fluent 2021 R2 (Ansys Inc., Canonsburg, PA, USA). All simulation tasks were performed on a local HP Z840 workstation (Intel^®^ Xeon^®^ Processor E5−2687W v4 with dual processors, 24 cores, 48 threads, and 128 GB RAM), and it required ~28 h to finish the simulation with time step size 5 × 10^−4^ s for one pulsatile period, i.e., T = 1.0 s. Three cardiac periods were simulated for each case, and the results were analyzed based on the third period. The Semi-Implicit Method for the Pressure Linked Equations (SIMPLE) algorithm was employed for the pressure-velocity coupling, and the least-squares cell-based scheme was applied to calculate the cell gradient. The second-order scheme was used for the discretization of pressure and momentum. Convergence is defined for continuity and momentum equations with the residual smaller than 1 × 10^−4^.

## 3. Results and Discussion

### 3.1. Effects of Pulsatile Flow Rate

To investigate the effects of pulsatile flow rate on hemodynamic characteristics in CAs, CFD simulation results are compared under three different pulsatile flow rates (e.g., PFR−I, PFR−II, and PFR−III) (see Figure 3) in two patient-specific ICASA models (i.e., ICASA−1 and ICASA−2), respectively. The relationships of three designated pulsatile flow rates are represented by variances of the volumetric flowrates (i.e., $q_{\mathrm{PFR}-I}$ = $0.57q_{\mathrm{PFR}-\mathrm{II}}$ and $q_{\mathrm{PFR}-\mathrm{III}}$ = $1.47q_{\mathrm{PFR}-\mathrm{II}}$) in the ICA inlet (see Figure 3 and Section 2.6). Overall, it can be found in the elliptically highlighted regions (red solid lines) of all simulated cases with different pulsatile flow rates and shunt ratios (see Figure 4, Figure 5 and Figure 6) that partial aneurysm neck regions suffer relatively larger WSS than other local regions in both investigated ICASA models, as high as >300 Pa under the largest pulsatile flow rate (i.e., PFR−III) at the peak systole point (i.e., t_2_ = 0.22 s) in the ICASA−2 model with q_A1_:q_A2_ = 75:25 (see Table 2). This is due to the direct impingements by the flow stream since part of the flow is redirected into the aneurysm sac from the ICA flow stream, which was visualized by the flow streamlines shown in Figure 7. Furthermore, relatively high OSI distributions can be observed in aneurysmal neck regions where flow separations occur when the bloodstream is approaching those regions (see Figure 8). These phenomena indicate that the local neck region in the cerebral aneurysms (highlighted in Figure 4, Figure 5 and Figure 6) may have higher potential risks of forming small or secondary aneurysms, which aligns with the hypothesis that the large WSS integrated with a positive WSS gradient could trigger a mural-cell-mediated pathway that could be allied with the generation, growth, and rupture of small or secondary bleb aneurysm phenotypes [16,63]. On the other hand, the low WSS and high OSI circumstances can be discovered in the highlighted regions with circled dashed lines in Figure 4, Figure 5, Figure 6 and Figure 8 under all three investigated PFR conditions. These luminal surfaces could have a higher probability of becoming enlarged/ruptured positions in the aneurysmal sacs by inducing inflammatory-cell-mediated destructive remodeling [16].

Figure 4a–c shows noticeable differences among the PFRs in calculations of WSS in the aneurysm of the ICASA−1 model under selected instants, and similar phenomena were also found in the ICASA−2 model (see Figure 5 and Figure 6). It is not surprising that the larger pulsatile flow rate (i.e., PFR−III) leads to a larger WSS than the smaller pulsatile flow rates (i.e., PFR−I and PFR−II) in all simulated cases at the same bifurcated shunt ratio (see Figure 4, Figure 5 and Figure 6). Table 2 and Table 3 show the same phenomena with more detailed WSS calculations (i.e., maximum instantaneous WSS and minimum instantaneous WSS in correspond axis directions in S1 and S2, and surface-averaged instantaneous $\bar{WSS}$ in R1 and R2) at representative time instants. Such an increase in WSS for the patients with a large pulsatile flow rate could facilitate the higher probability of secondary aneurysm generation and/or thin-wall symptoms in the existing aneurysm since the ICA wall tension/deformation may increase and then overcome thresholds of the wall tissue compliance, which is consistent with the perspectives from previous studies [64,65]. Quantitatively, at time instants t_1_ = 0.12 s, t_1_ = 0.22 s, t_3_ = 0.40 s, and t_4_ = 0.80 s, the surface-averaged wall shear stresses instantaneous $\bar{WSS}$ under PFR−II and PFR−III are about one and three times larger than the instantaneous $\bar{WSS}$ under PFR−I at regions near the aneurysmal sac (R1) of the ICASA−1 model, respectively (see Table 3). Similar ratios can also be observed at the same instant in the R2 of the ICASA−2 model shown in Table 3. These comparable discoveries can be explained well with the classic fluid dynamic theory [66] that the larger flow rate can lead to a larger velocity gradient (see Equation (6)) within the boundary layer near the arterial wall surface. The increased near-surface velocity gradient in these regions gives rise to larger viscous shear stresses on the aneurysmal wall. These phenomena can be further explained directly based on the comparisons in velocity profiles in the ICASA models. At t_2_ = 0.22 s, the nondimensionalized velocity magnitude $V^{*}$ at the extracted lines (i.e., lines a, b, and c in ICASA−1, and lines d and e in ICASA−2) are shown in Figure 9a,b, respectively. As shown by the comparisons made among the three velocity profiles near the artery wall that are enlarged and shown in Figure 9, the higher pulsatile flow rate results in a steeper slope (i.e., larger velocity gradient) and leads to a larger WSS on the selected aneurysmal wall, along with stronger wall impingements by the bulkier blood flows (see Figure 7). The highlighted circled regions in distributions of velocity vectors in Figure 10 also support this explanation. Conversely, a lower blood flow rate results in less blood streams entering the aneurysmal sac (see Figure 10a,d,g) and thus smaller velocity gradients on the arterial wall (see Figure 9a,b). Further, such an explanation reveals the differences in comparisons of instantaneous $\bar{WSS}$ at t_1_ = 0.12 s among PFR−I, PFR−II, and PFR−III are much smaller than other designated time instants, e.g., t_2_, t_3_, and t_4_ (see Figure 3).

It should not be noted that although the lower pulsatile flow rate may influence the aneurysm sac insignificantly when considering the low WSS solely (see Figure 4a, Figure 5a and Figure 6a), the OSI is still high on a larger surface area at some local regions, i.e., especially on the fundus of the aneurysm dome (see Figure 8a,d,g), under such small pulsatile flow rate (i.e., PFR−I) than its counterparts under large pulsatile flow rates (e.g., PFR−II and PFR−III) (see Figure 8b,c,e–i). Low WSS and high OSI are well-known prevalent hypotheses to trigger the growth and rupture of cerebral aneurysms via malfunctioning of the endothelial surface to produce nitric oxide, increasing endothelial permeability, and consequently promoting inflammatory cell infiltration [67,68,69]. Thus, reducing OSI in some local regions (highlighted in Figure 8) for patients with a lower pulsatile flow rate should be a critical concern when clinicians consider treatments to decrease the risks of aneurysmal sac rupture. For instance, stent and coil treatments may be executed in the arteries and aneurysms to alter blood flow patterns and reduce OSI.

With respect to the PFR effects on blood pressure, the highlighted regions in Figure 11a–c and Table 4 manifest that larger PFR (i.e., PFR−III) leads to relatively higher TAP on aneurysmal walls since more blood is redirected into the aneurysm sac from the ICA regime, which causes higher velocity impingement on the sac wall (see Figure 7a–c and Figure 10a–c), and then leads to higher blood pressure than the other two smaller PFRs (i.e., PFR−I and PFR−II) at the same instants. The high blood pressure associated with the larger PFR can be a potential risk factor for the pathophysiology of a cerebral aneurysm [57] since the induced hemodynamic stress and inflammation by high blood pressure could result in the growth and rupture of CAs by causing arterial wall damage and dilation [58,59]. Nevertheless, it can be interestingly found in Figure 11a–i that the variance of PFR has no extremely significant impact on TAP distributions on the aneurysmal dome wall in both ICASA models since, compared to the static pressure, the contribution from dynamic pressure is insignificant to affect the total blood pressure intrinsically. Specifically, TAP distributions vary from 13,310 to 15,050 Pa on the aneurysmal wall in ICASA−1 and 13,097 to 13,202 Pa on the aneurysmal wall in ICASA−2, respectively. Some local regions close to the lower portions of the aneurysmal neck (highlighted with dash lines) in ICASA−2 register relatively higher TAP (i.e., 13,202 to 13,411 Pa), which is because the local velocity at these regions is higher than other regions in the sac, where TAP is distributed relatively uniformly (i.e., nearly 13,000 Pa) due to smaller velocity differences under the three PFRs (see Figure 7d–i and Figure 10d–i).

### 3.2. Effects of Shunt Ratios in Bifurcated Distal Arteries

To analyze hemodynamic characteristics in CAs due to different shunt ratios in the bifurcated A1 and A2, CFD results were compared using two shunt ratios of q_A1_:q_A2_, e.g., 75:25 and 64:36, in the patient-specific ICASA−2 model. Specifically, the WSS (see Figure 5 and Figure 7), velocity streamlines (see Figure 7d–i), OSI (see Figure 8d–i), nondimensionalized velocity profiles on selected lines (see Figure 9b), velocity vectors in selected slices (see Figure 10d–i), and TAP distributions (see Figure 11d–i) were compared between the two shunt ratios (i.e., 75:25 and 64:36), respectively. Overall, it can be observed that the flow rates in A1 and A2 have minimal effects on the blood flow field in the aneurysmal sac. All of the above-mentioned comparable parameters (i.e., WSS, TAP, velocity profiles, and OSI) at the two ratios showed nearly identical distributions in the cerebral aneurysm sac under the same PFR. Such limited influence on flow patterns in the aneurysmal dome by the variance of shunt ratios in bifurcated distal arteries is because of the fact that the distance from the aneurysmal sac to the bifurcation point is too far to alter the blood flow patterns in the dome. The only minor difference in the averaged wall shear stress $\bar{WSS}$ and TAP was observed in the studied aneurysmal region, R2. The $\bar{WSS}$ under q_A1_:q_A2_ = 75:25 was a little bit smaller than the counterparts under q_A1_:q_A2_ = 64:36 at designated instants (see Table 4), which is due to the fact that the integral scope to compute $\bar{WSS}$ not only contains the aneurysmal sac, but also takes partial arterial walls of ICA, A1, and A2 into consideration (see Figure 10 and Figure 11). This fact can also explain the minor differences in *TAP* under the two shunt ratios (see Table 4).

To investigate more specifically the critical distance from the bifurcated arteries (i.e., A1 and A2) to the aneurysmal sac beyond which the shunt ratio variance can influence the flow patterns in the brain aneurysm model (i.e., ICASA−2), the flow details in this region have been visualized in the designated plane C−C′ in Figure 12 at the time instant of t_2_ = 0.22 s under PFR−II. Figure 12a,b show velocity magnitude contours under the two-shunt ratios, i.e., q_A1_:q_A2_ = 75:25 and q_A1_:q_A2_ = 64:36, respectively. It can be found that the red-colored rectangular region has the same flow field under both shunt ratios, in accordance with the nondimensionalized velocity profiles along the selected line f (see Figure 12c). The visualized velocity profiles along the designated line g (colored with pink) also support the observation that *V** profiles are identical under the two shunt ratios shown in Figure 12c. The flow patterns begin to differ in the two ratios when $L^{*}$ is smaller than 0.405 (see Figure 12c). It implies that only the aneurysmal sac is formed close enough to the bifurcated arteries, i.e., $0<L^{*}\leq 0.405$ in the ICASA−2 model, the differences in shunt ratios in bifurcated distal arteries can show a noticeable impact on the blood transport behaviors in the aneurysmal sacs. The discoveries in this study may help clinicians obtain more accurate information when the treatments are attempted for patients with CA issues that are closely approaching the bifurcated arteries. Nevertheless, it should be mentioned that the critical value $L^{*}$ may vary with the PFR conditions, and the qualitative and quantitative statistic investigations on the effects of bifurcated blood distributions on WSS and OSI in the aneurysm still need to be conducted based on the patient-specific cerebral aneurysms (e.g., close enough to the bifurcated arteries) in future studies.

### 3.3. Effects of Transitional Pulsatile Blood Flow

During a single cardiac pulse period, i.e., *T* = 1.0 s, the WSS on arterial walls and blood flow patterns in the aneurysmal sac vary with the flow rate. As shown in Figure 3, the blood flow rate drops slowly from the beginning to 0.14 s, then increases sharply to the maximum at t_2_ = 0.22 s, followed by a steep decrease and an oscillation to the second largest peak from 0.22 s to 0.44 s, then declines gradually till one cardiac period is concluded. Meanwhile, in one cardiac period, the induced WSS in the aneurysm sac presents a similar trend as the transient pulsatile flow rate waveform, showing that the larger volumetric flow rate induces a larger velocity gradient, thus leading to a larger WSS on the aneurysmal sac wall. Specifically, the maximum WSS appears at t_2_ = 0.22 s with 587.266 Pa under PFR−III with q_A1_:q_A2_ = 75:25 (see Figure 6 and Table 2) in the ICASA−2 model, while the minimum WSS is 15.27 Pa under PFR−I with qA1:qA2 = 75:25 in ICASA−1 model. However, comparing $\bar{WSS}$, the maximum value of 82.072 Pa appears at the designated region (e.g., R1 in ICASA−1) at the peak systole (i.e., t_2_ = 0.22 s) (see Table 3). This is because of the selected integral region for calculations of the $\bar{WSS}$ and it has been explained in Section 3.2. More specifically, for instance, during one cardiac period in the two investigated ICASA models under PFR−II and q_A1_:q_A2_ = 75:25 (see Table 2), the maximum WSS are 43.36, 267.12, 122.73, and 59.10 Pa on the sac of ICASA−1 (S1) and 86.16, 343.39, 213.20, and 107.15 Pa on the sac of the ICASA−2 (S2) under the three mass flow rates at the representative time instants (i.e., t_1_ = 0.14 s, t_2_ = 0.22 s, t_3_ = 0.40 s, and t_4_ = 0.80 s), which are 3.18 × 10^−3^, 9.93 × 10^−3^, 6.48 × 10^−3^, and 3.89 × 10^−3^ kg/s, respectively. As the mass flow rate increases about two times from the flowing time t_1_ to t_2_, its corresponding maximum WSS and $\bar{WSS}$ increase over four times in ICASA−1 and three times in ICASA−2 in all studied PFRs, respectively. Such results are observed mainly because the velocity gradient increases more intensively as the blood flow rate increases during one pulse period. These findings may provide guidelines for consulting patients with CAs to avoid aggressive sports and exercises since the blood flow rate during maximum exercise may increase up to four times the value for the rest state [70]. The significantly increased flow rate can cause a much higher WSS on the aneurysmal wall and then pose potential risks to facilitate the aneurysm growth or rupture.

To present the blood flow characteristics, the visualized velocity contours and flow streamlines in the plane of Y = −0.180 m in the ICASA−1 model and in the plane of X = 0.007 m in the ICASA−2 model have been presented in Figure 13a,b at the representative time instants (i.e., t_1_ = 0.14 s, t_2_ = 0.22 s, t_3_ = 0.40 s, and t_4_ = 0.80 s), separately. The CFD results show that the largest WSS is always located in the sac neck region (highlighted in Figure 13a,b), which is due to the strongest impingement by the bloodstream onto the aneurysmal sac in both ICASA models, shown in Figure 13c,d. Figure 13a,b show that the vortex plays a major role in the temporal and spatial distribution of the WSS on the sac wall, varying significantly along with varying flow rates during the cardiac period. The unsteady vortex in the aneurysmal sac increases velocity fluctuations, which not only leads to high WSS (see Figure 13c,d), but also causes high OSI, which has been discussed in Section 3.1. The findings in this study are consistent with the previous study that tracking the vortex formation and growth is a crucial step in analyzing hemodynamic factors on the pathophysiology of cerebral aneurysms [71]. Moreover, it is suggested that the researchers should avoid using a non-pulsatile blood flow condition to conduct CFD simulations to investigate hemodynamic characteristics in cerebral aneurysms since it may cause significant differences between non-pulsatile and pulsatile conditions when predicting the flow characteristics in CAs.

## 4. Conclusions

In this study, the effects of pulsatile flow rates, shunt ratios in bifurcated distal arteries, and transient cardiac pulsatile blood flow on hemodynamic transport behaviors in the ICASA−1 and ICASA−2 models were investigated using CFD. Specifically, the main conclusions are summarized as follows:

The pulsatile flow rate has a significant impact on hemodynamic characteristics in cerebral aneurysms. Larger pulsatile flow rates lead to higher WSS in the aneurysmal region, which may increase the risk of forming small/secondary aneurysms. Although aneurysmal artery walls may suffer lower WSS under a lower pulsatile flow rate, the high OSI distributed in local regions may affect the growth and rupture of cerebral aneurysms.The variances of shunt ratios in bifurcated distal arteries have no significant impact on the hemodynamic behaviors in the aneurysmal sac because the distal bifurcated location is not close enough to the aneurysm sac in the ICASA−2 model. We concede that more specific qualitative and quantitative investigations of the effects of bifurcated shunt ratios on flow characteristics in the aneurysmal sac using patient-specific cerebral aneurysms are still needed.A higher PFR can contribute more to the pressure increase in the ICASA−1 dome due to the stronger impingement by the splitting bloodstream, while the variances of PFR and shunt ratio in the bifurcated distal arteries have rare impacts on the dome of the ICASA−2 model since only a small part of the bloodstream will be redirected into the sac.The regions in the neck of the aneurysmal sac with higher WSS may lead to a high incidence of small/secondary aneurysm generation under all studied pulsatile flow rates and bifurcated shunt ratios. Moreover, some local luminal surfaces on the aneurysmal dome could have a higher probability of enlarging/rupturing, given the evidence of relatively high OSI and low WSS features.During one pulse period, the blood flow at the systolic peak can influence the hemodynamic patterns (i.e., WSS and vortex) considerably more than other time instants. The slope of the increase of WSS is beyond the slope of the increase of the blood flow rate, and this phenomenon is more apparent under a smaller PFR.

In conclusion, the findings in this work can contribute to the intracranial aneurysm community’s knowledge by providing a better understanding of blood flow patterns in the aneurysmal sac and the effects of different patient-specific pulsatile flow rates and shunt ratios in bifurcated distal arteries.

## 5. Limitations and Future Work

Blood is a non-Newtonian fluid with shear-thinning properties that were not considered in this study. Our following research will simulate the hemodynamic characteristics in the patient-specific cerebral aneurysm models using particle image velocimetry (PIV) measurements and computational fluid dynamics modeling, integrated with non-Newtonian blood properties. Furthermore, this study did not simulate the arterial wall deformations coupled with the hemodynamic patterns explicitly. Accordingly, our long-term goal is to build a realistic in silico model to conduct statistical analysis of the hemodynamic factors on the pathophysiology of cerebral aneurysms (i.e., nearly 100 patient-specific aneurysmal models) using a two-way fluid-solid interaction (FSI) manner, i.e., the deformation effects between the cerebral artery wall and the realistic non-Newtonian blood. Based on the statistical investigations, more hemodynamic information in CAs will be summarized and identified to assist clinical applications, i.e., diagnosis and treatment.

## Figures and Tables

**Figure 1 bioengineering-09-00326-f001:**
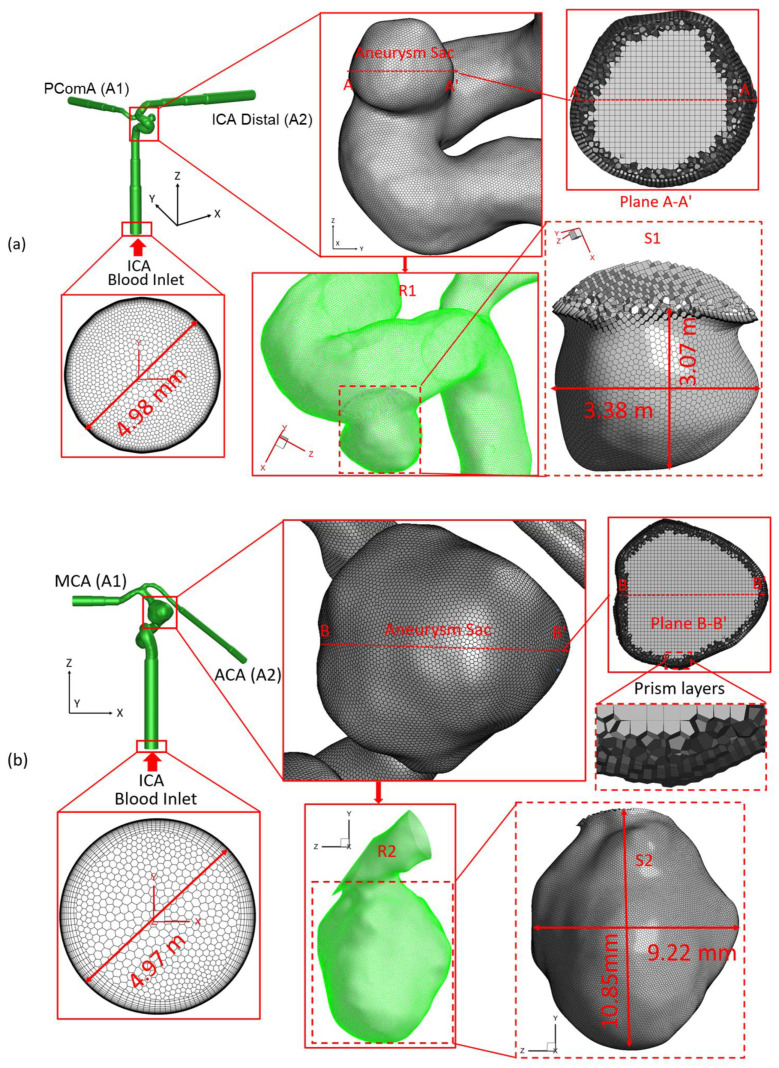
Schematic of the computational domain with hybrid mesh details in ICASA models: (**a**) ICASA−1 and (**b**) ICASA−2.

**Figure 2 bioengineering-09-00326-f002:**
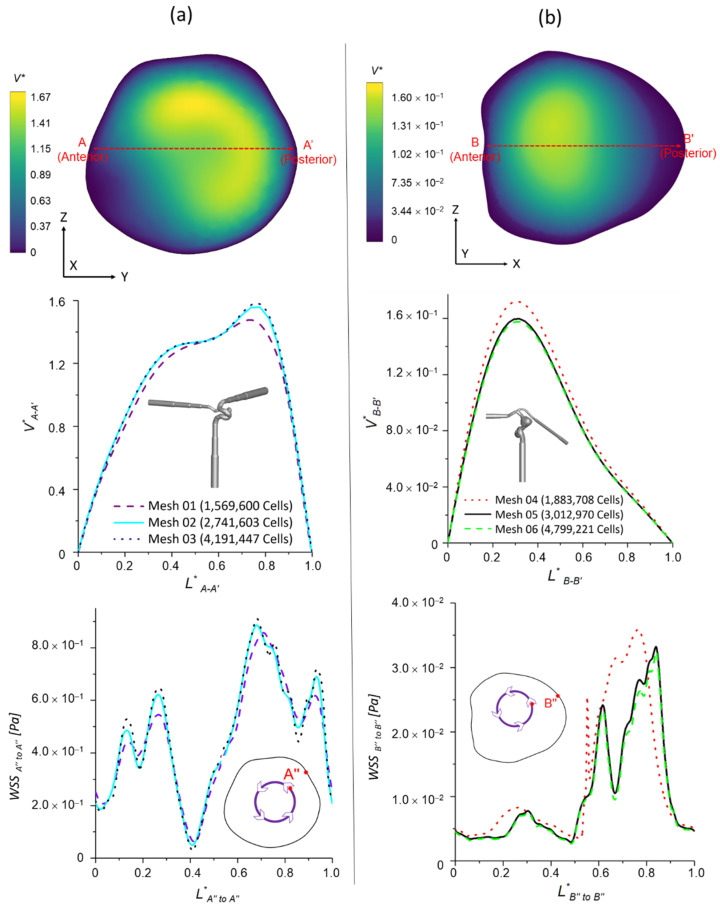
Mesh independence tests for the two CA models: (**a**) ICASA−1 and (**b**) ICASA−2.

**Figure 3 bioengineering-09-00326-f003:**
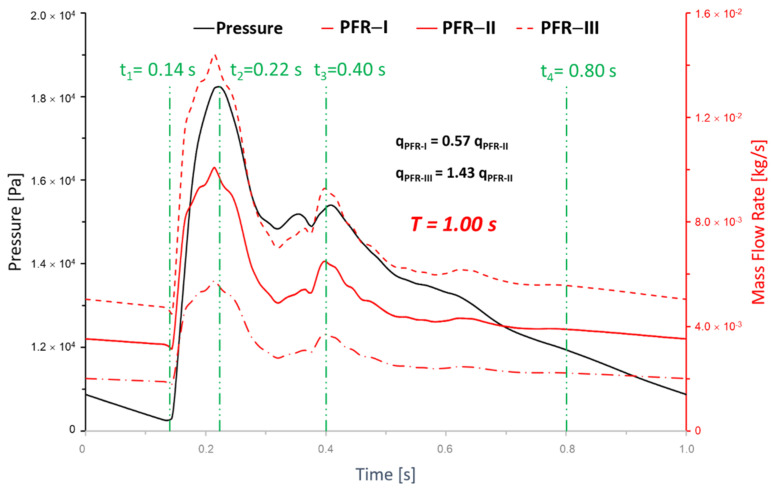
Transient pulsatile flow rate boundary conditions at the internal carotid artery (ICA) inlet.

**Figure 4 bioengineering-09-00326-f004:**
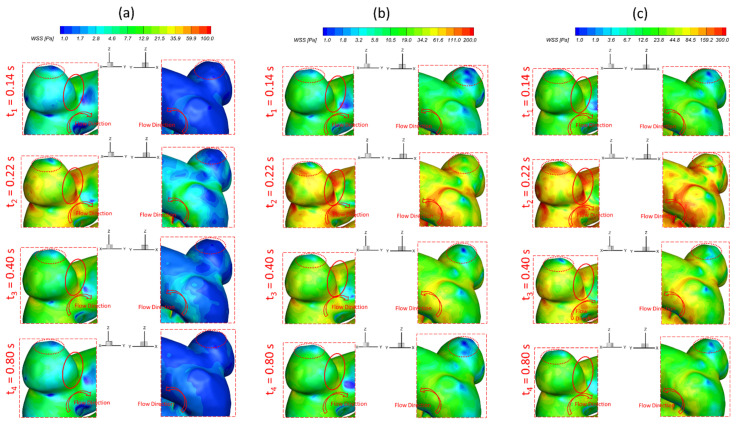
Visualized instantaneous WSS distributions on selected region (R1) of ICASA−1 model at selected instants and shunt ratio of q_A1_:q_A2_ = 75:25 under corresponded PFRs: (**a**) PFR−I, (**b**) PFR−II, and (**c**) PFR−III.

**Figure 5 bioengineering-09-00326-f005:**
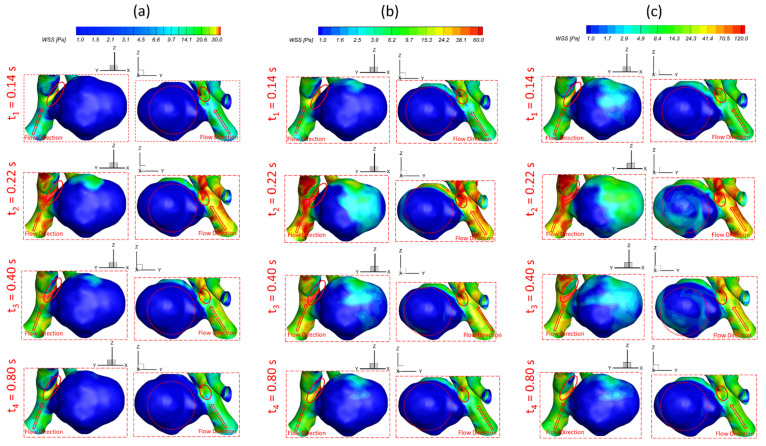
Visualized instantaneous WSS distributions on selected region (R2) of ICASA−2 model at selected instants and shunt ratio of q_A1_:q_A2_ = 75:25 under corresponded PFRs: (**a**) PFR−I, (**b**) PFR−II, and (**c**) PFR−III.

**Figure 6 bioengineering-09-00326-f006:**
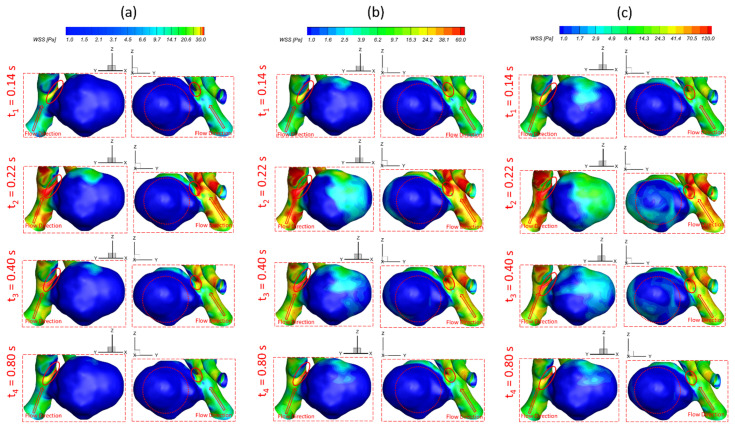
Visualized instantaneous WSS distributions on selected region (R2) of ICASA−2 model at selected instants and shunt ratio of q_A1_:q_A2_ = 64:36 under corresponded PFRs: (**a**) PFR−I, (**b**) PFR−II, and (**c**) PFR−III.

**Figure 7 bioengineering-09-00326-f007:**
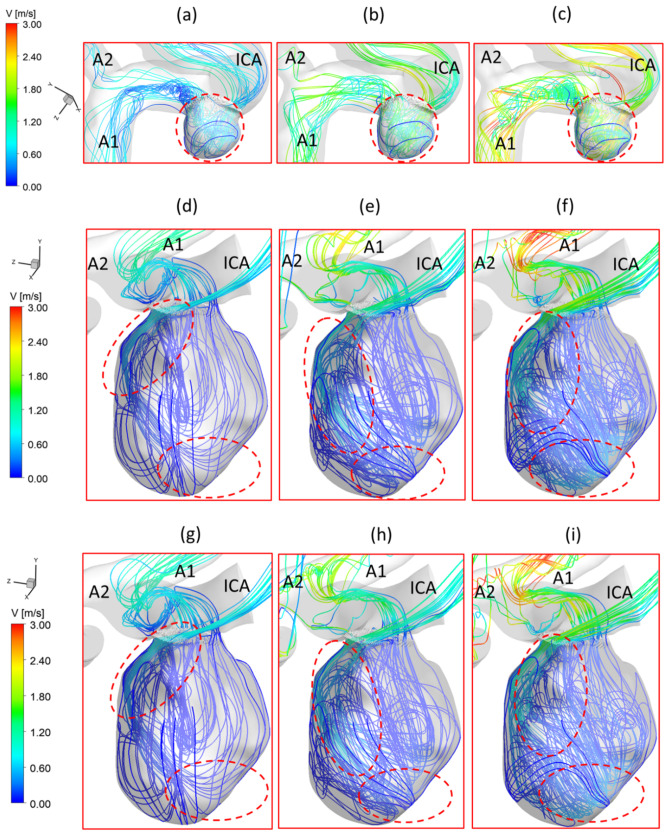
Visualized flow streamlines in ICASA−1 (S1) and ICASA−2 (S2) models under different conditions at t_2_ = 0.13 s: (**a**) ICASA−1, q_A1_:q_A2_ = 75:25, and PFR−I, (**b**) ICASA−1, q_A1_:q_A2_ = 75:25, and PFR−II, (**c**) ICASA−1, q_A1_:q_A2_ = 75:25, and PFR−III, (**d**) ICASA−2, q_A1_:q_A2_ = 75:25, and PFR−I, (**e**) ICASA−2, q_A1_:q_A2_ = 75:25, and PFR−II, (**f**) ICASA−2, q_A1_:q_A2_ = 75:25, and PFR−III, (**g**) ICASA−2, q_A1_:q_A2_ = 64:36, and PFR−I, (**h**) ICASA−2, q_A1_:q_A2_ = 64:36, and PFR−II, and (**i**) ICASA−2, q_A1_:q_A2_ = 64:36, and PFR−III.

**Figure 8 bioengineering-09-00326-f008:**
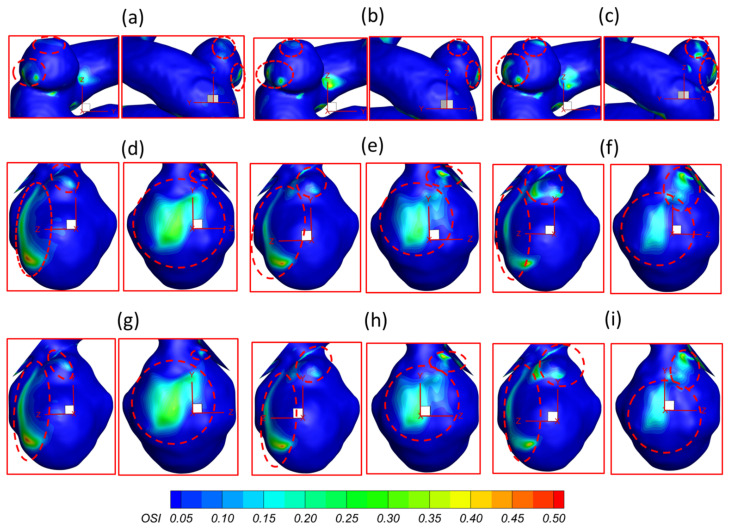
OSI distributions in ICASA−1 and ICASA−2 models under different conditions: (**a**) ICASA−1, q_A1_:q_A2_ = 75:25, and PFR−I, (**b**) ICASA−1, q_A1_:q_A2_ = 75:25, and PFR−II, (**c**) ICASA−1, q_A1_:q_A2_ = 75:25, and PFR−III, (**d**) ICASA−2, q_A1_:q_A2_ = 75:25, and PFR−I, (**e**) ICASA−2, q_A1_:q_A2_ = 75:25, and PFR−II, (**f**) ICASA−2, q_A1_:q_A2_ = 75:25, and PFR−III, (**g**) ICASA−2, q_A1_:q_A2_ = 64:36, and PFR−I, (**h**) ICASA−2, q_A1_:q_A2_ = 64:36, and PFR−II, and (**i**) ICASA−2, q_A1_:q_A2_ = 64:36, and PFR−III.

**Figure 9 bioengineering-09-00326-f009:**
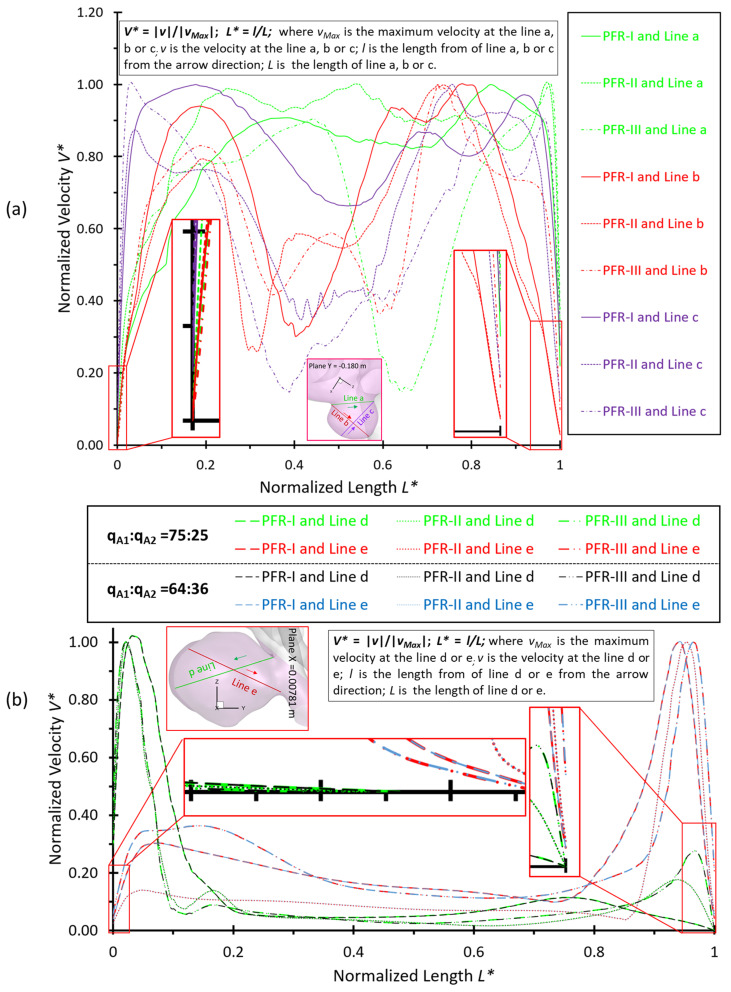
Nondimensionalized velocity profiles *V** at the designated cross lines across the aneurysmal sac at t_2_ = 0.22 s in ICASA models: (**a**) ICASA−1 and (**b**) ICASA−2.

**Figure 10 bioengineering-09-00326-f010:**
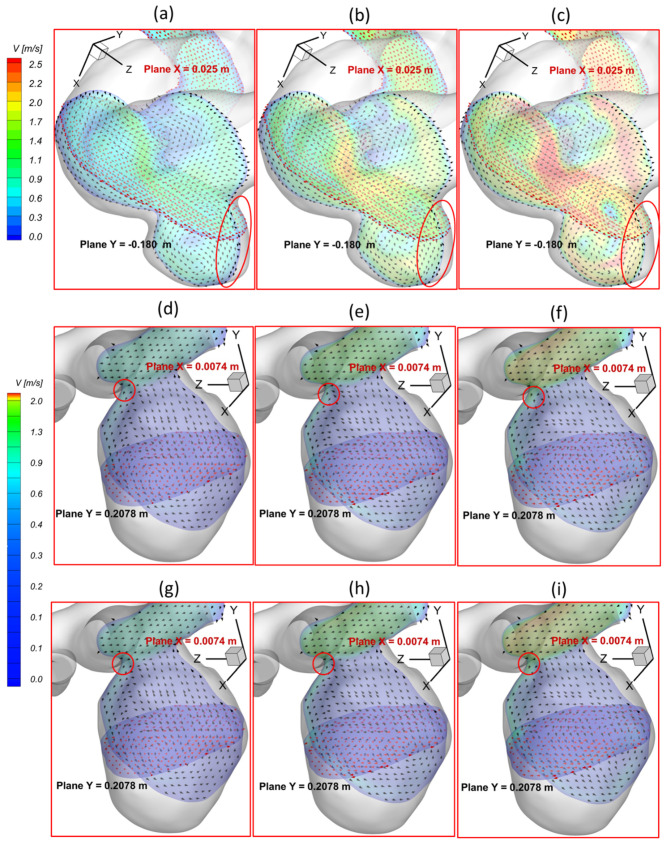
Visualized velocity profiles at t_2_ = 0.22 s in extracted planes of ICASA models: (**a**) ICASA−1, q_A1_:q_A2_ = 75:25, and PFR−I, (**b**) ICASA−1, q_A1_:q_A2_ = 75:25, and PFR−II, (**c**) ICASA−1, q_A1_:q_A2_ = 75:25, and PFR−III, (**d**) ICASA−2, q_A1_:q_A2_ = 75:25, and PFR−I, (**e**) ICASA−2, q_A1_:q_A2_ = 75:25, and PFR−II, (**f**) ICASA−2, q_A1_:q_A2_ = 75:25, and PFR−III, (**g**) ICASA−2, q_A1_:q_A2_ = 64:36, and PFR−I, (**h**) ICASA−2, q_A1_:q_A2_ = 64:36, and PFR−II, and (**i**) ICASA−2, q_A1_:q_A2_ = 64:36, and PFR−III.

**Figure 11 bioengineering-09-00326-f011:**
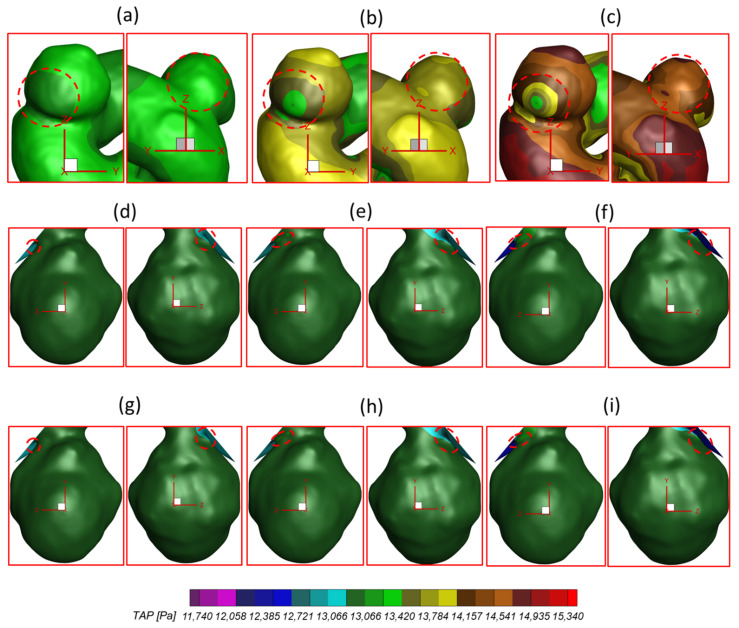
TAP distributions on the arterial walls of ICASA models: (**a**) ICASA−1, q_A1_:q_A2_ = 75:25, and PFR−I, (**b**) ICASA−1, q_A1_:q_A2_ = 75:25, and PFR−II, (**c**) ICASA−1, q_A1_:q_A2_ = 75:25, and PFR−III, (**d**) ICASA−2, q_A1_:q_A2_ = 75:25, and PFR−I, (**e**) ICASA−2, q_A1_:q_A2_ = 75:25, and PFR−II, (**f**) ICASA−2, q_A1_:q_A2_ = 75:25, and PFR−III, (**g**) ICASA−2, q_A1_:q_A2_ = 64:36, and PFR−I, (**h**) ICASA−2, q_A1_:q_A2_ = 64:36, and PFR−II, and (**i**) ICASA−2, q_A1_:q_A2_ = 64:36, and PFR−III.

**Figure 12 bioengineering-09-00326-f012:**
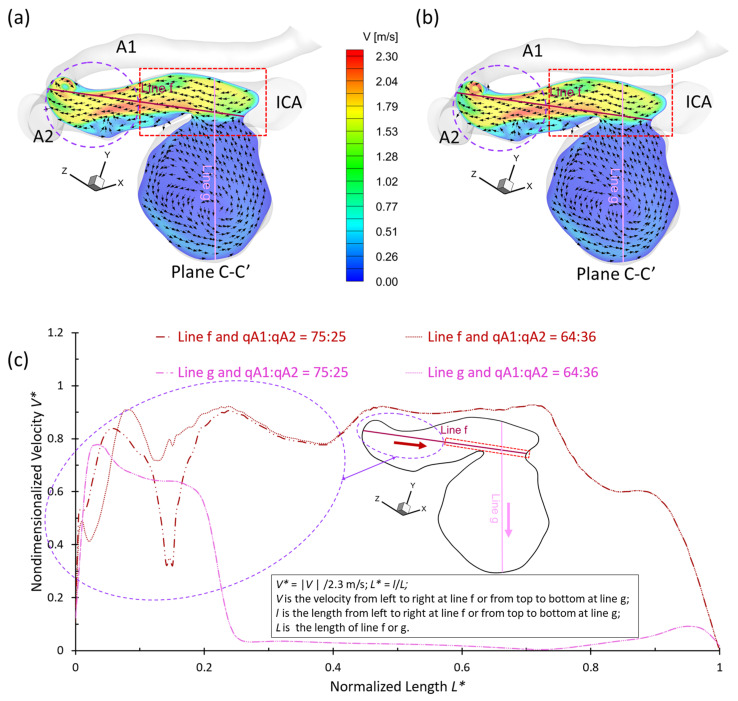
Comparisons of velocity profiles in the extracted plane C−C′ and velocity distributions in selected lines f and g in ICASA−2 model under different bifurcated shunt ratios: (**a**) velocity profiles at plane C−C′ under q_A1_:q_A2_ = 75:25, (**b**) velocity profiles at plane C−C′ under q_A1_:q_A2_ = 64:36, and (**c**) nondimensionalized velocity profiles at lines g and f.

**Figure 13 bioengineering-09-00326-f013:**
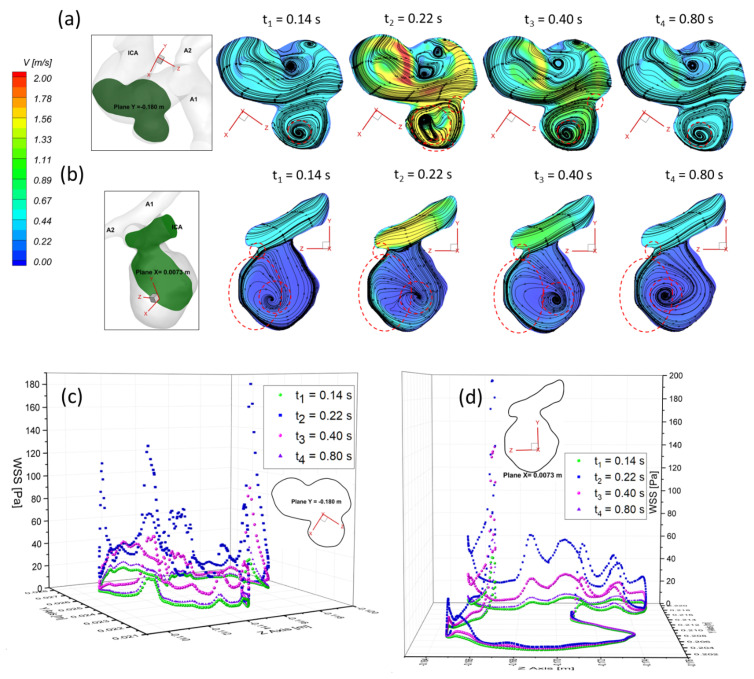
Velocity profiles and flow streamlines in selected planes and WSS distributions on the edges of selected planes in ICASA−1 and ICASA−2 models at selected instants: (**a**) velocity profiles and streamlines in plane Y = −0.180 m of ICASA−1 model, (**b**) velocity profiles and streamlines in plane X = 0.0073 m of ICASA−2 model, (**c**) WSS distributions at the edges of plane Y = −0.180 m of ICASA−1 model, and (**d**) WSS distributions at the edges of plane X = 0.0205 m of ICASA−2.

**Table 1 bioengineering-09-00326-t001:** Mesh details in mesh independence tests for ICASA models.

Mesh	Minimum Size (mm)	Face Elements	Face-Maximum Skewness	Volume Elements	Volume-Maximum Skewness	Prism Layers	First Prism Layer Height (m)	Peel Layers	Size Growth Rate
Mesh 01	3.5 × 10^−4^	31,930	0.45	1,569,600	0.88	15	2.2 × 10^−^^6^	3	1.05
Mesh 02 (Final)	3.0 × 10^−4^	99,170	0.47	2,741,603	0.89	25	1.8 × 10^−6^		
Mesh 03	2.5 × 10^−4^	271,511	0.44	4,191,447	0.88	30	1.5 × 10^−6^		
Mesh 04	4.0 × 10^−4^	30,589	0.29	1,883,708	0.89	15	2.0 × 10^−6^		
Mesh 05 (Final)	3.5 × 10^−4^	126,896	0.42	3,012,970	0.87	20	1.5 × 10^−6^		
Mesh 06	3.0 × 10^−4^	166,901	0.38	4,799,221	0.86	25	1.0 × 10^−6^		

**Table 2 bioengineering-09-00326-t002:** Maximum instantaneous WSS (Pa) in the aneurysm sacs (S1 and S2) under the bifurcated shunt ratio q_A1_:q_A2_ = 75:25 and different PFRs at designated time instants.

Aneurysmal Sac	PFR	WSSCategories	Time Instant (s)			
			t_1_ = 0.14	t_2_ = 0.22	t_3_ = 0.40	t_4_ = 0.80
ICASA−1 (S1)	PFR−I	$WSS_{Max}$	15.2711	173.537	40.8334	19.869
		$WSS_{x\_Max}$	−9.98615	−81.9959	−34.0993	−13.9671
		$WSS_{y\_Max}$	13.3374	73.5389	36.5139	17.596
		$WSS_{z\_Max}$	12.1799	91.7417	32.5699	16.5839
	PFR−II	$WSS_{Max}$	43.3625	267.123	122.726	59.102
		$WSS_{x\_Max}$	−29.8159	−220.707	−96.8994	−41.3871
		$WSS_{y\_Max}$	32.3394	191.372	90.6823	42.4013
		$WSS_{z\_Max}$	36.2225	231.593	108.234	50.3479
	PFR−III	$WSS_{Max}$	80.9053	431.082	231.061	108.233
		$WSS_{x\_Max}$	−58.853	−361.069	−194.217	−82.4116
		$WSS_{y\_Max}$	56.5928	249.25	154.078	74.6838
		$WSS_{z\_Max}$	71.5142	391.926	206.949	98.4345
ICASA−2 (S2)	PFR−I	$WSS_{Max}$	34.0782	169.003	92.7069	45.3113
		$WSS_{x\_Max}$	−22.7832	−87.9672	−54.4009	−28.8848
		$WSS_{y\_Max}$	25.3666	116.035	68.6824	32.9661
		$WSS_{z\_Max}$	24.1866	135.185	69.7968	32.8628
	PFR−II	$WSS_{Max}$	86.1642	343.389	213.202	107.149
		$WSS_{x\_Max}$	−52.2246	−146.942	−107.226	−60.6988
		$WSS_{y\_Max}$	60.534	234.625	153.317	75.4735
		$WSS_{z\_Max}$	63.318	269.778	174.383	84.017
	PFR−III	$WSS_{Max}$	138.585	587.266	321.377	175.407
		$WSS_{x\_Max}$	−77.2774	−276.796	−130.745	−82.6379
		$WSS_{y\_Max}$	98.1775	392.104	214.711	117.631
		$WSS_{z\_Max}$	109.770	438.358	251.712	140.801

**Table 3 bioengineering-09-00326-t003:** $\bar{WSS}$ (Pa) in aneurysmal regions (R1 and R2) under selected instants and boundary conditions.

Selected Region	q_A1_:q_A2_	Pulsatile Flow Rate	Time Instant (s)			
			t_1_ = 0.14	t_2_ = 0.22	t_3_ = 0.40	t_4_ = 0.80
			$\bar{WSS}\;(\mathrm{Pa})$			
ICASA−1 (R1)	75:25	PFR−I	3.783	19.230	9.586	4.933
		PFR−II	8.752	46.121	22.782	11.310
		PFR−III	15.105	82.072	39.916	19.736
ICASA−2 (R2)	75:25	PFR−I	2.359	11.390	6.264	3.061
		PFR−II	5.292	26.509	13.909	6.900
		PFR−III	8.989	44.884	23.743	11.518
	64:36	PFR−I	2.543	11.863	6.402	3.282
		PFR−II	5.594	27.110	14.258	7.180
		PFR−III	9.306	46.224	23.961	11.729

**Table 4 bioengineering-09-00326-t004:** TAP (Pa) in aneurysmal regions (R1 and R2) under different PFRs and shunt ratios.

Selected Region	q_A1_:q_A2_	Pulsatile Flow Rate	TAP (Pa)	
			Minimum	Maximum
ICASA−1 (R1)	75:25	PFR−I	13,114.6	13,613.0
		PFR−II	12,973.9	14,301.0
		PFR−III	12,732.1	15,324.9
ICASA−2(R2)	75:25	PFR−I	12,884.3	13,262.5
		PFR−II	12,399.9	13,429.7
		PFR−III	11,742.8	13,613.9
	64:36	PFR−I	12,884.4	13,263.0
		PFR−II	12,400.0	13,430.0
		PFR−III	11,743.3	13,614.2

## Data Availability

Data available on request due to restrictions e.g., privacy or ethical.

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
