# Peer review of "Effects of Pulsatile Flow Rate and Shunt Ratio in Bifurcated Distal Arteries on Hemodynamic Characteristics Involved in Two Patient-Specific Internal Carotid Artery Sidewall Aneurysms: A Numerical Study"

_bioengineering, 2022, doi:10.3390/bioengineering9070326_

Round 1

Reviewer 1 Report

Manuscript review of "Effects of Pulsatile Flow Rate and Shunt Ratio in Bifurcated Distal Arteries on Hemodynamic Characteristics Involved in Two Patient-Specific Internal Carotid Artery Sidewall Aneurysms: A Numerical Study"

The authors investigate numerically the effect of inflow and outflow splitting in two models of cerebral aneurysm. They change only the magnitude of the input flux but not the shape, so it is expected that the WSS and the OSI change proportionally with this variation. The effects of flow division are not important since the outlets are far from the aneurysm.

In figures 9 and 12 not all cases are clearly visible, redo, or delete.

In addition, WSS graphs must be made, as in figures 4, 5 and 6 with details of the aneurysm and with a special scale to observe variations of WSS in the aneurysm, which is the area of interest. Also the same coments fore the TAP presented in figure 11.

In summary I do not see any relevance of this investigacion compared for example with:

The effect of inlet waveforms on computational hemodynamics of patient-specific intracranial aneurysms. J. Xiang a,b,c, A.H. Siddiqui a,c,d, H. Meng , Journal of Biomechanics, Volume 47, Issue 16, 18 December 2014, Pages 3882-3890.

If you take a new geometry with new bounbary conditions on cerebral aneurysms , you obtain new results,  but the authors must convince us that these new results are important and useful to the scientific community.

Reviewer 2 Report

The authors use computational fluid dynamics and internal carotid artery sidewall aneurysm (ICASA) models to delineate the blood flow pattern and the hemodynamic property in the aneurysmal sac. It provides insight into how pulsatile flow rate and shunt ratio can affect the hemodynamic characteristics of aneurysm and has a good point of therapeutic perspective in the future. However, there exist some issues needed to be clarified.

1.     The risk of aneurysm formation is different from the risk of aneurysm rupture. The hemodynamic characteristics described in the study might only reflect the causality of aneurysm formation and the resulting hemodynamic change. The result of the study does not provide more information about how to prevent overtreatment or undertreatment of unruptured intracranial aneurysm.

2.     The author proposes that the stents and coils should be placed in appropriate locations and directions from the standpoint of different flow patterns to decrease the risk of aneurysm rupture. However, the therapeutic effects of these interventions are stasis and spontaneous thrombosis. The analysis of hemodynamic characteristics in such circumstance is quite different from the real situation and the result might be of little clinical benefit.

3.     The risk of aneurysm formation and rupture are varied even in the same internal carotid artery system. Can author explain this phenomenon from your study result?

Reviewer 3 Report

The revised manuscript presents interesting study dedicated to CFD application for blood hemodynamic reconstruction in cerebral part of cardiac system. The authors present well-prepared and informative images together with a lot of details on blood flow. However, I have some comments listed below.

1.     Authors should rebuild the abstract. There is no information on used methodology.

2.     In the introduction part Authors should add some information dedicated to 3d technique applied in medical diagnosis/treatment. The authors should consider justifying the motivation of this study with recently published studies such as:

Polanczyk A, Wozniak T, Strzelecki M, Szubert W, Strzelecki M. Evaluating an algorithm for 3D reconstruction of blood vessels for further simulations of hemodynamic in human artery branches. Signal Processing - Algorithms, Architectures, Arrangements, and Applications Conference Proceedings, SPA. [Conference Paper]. 2016:5.

Polanczyk A, Strzelecki M, Wozniak T, Szubert W, Stefanczyk L. 3D Blood Vessels Reconstruction Based on Segmented CT Data for Further Simulations of Hemodynamic in Human Artery Branches. Foundations of Computing and Decision Sciences. [Conference Paper]. 2017;42(4):13.

3.      The authors analyzed two patients at the age of 73 and 35 years. The authors should justify why there was such age gap and how it could influence gathered results. It is well known that age determine vascular wall stiffness, blood pressure etc. hence it may change/influence obtained results.

4.     Pulsative flow applied as inlet boundary should be described in more details. It is not clear if it was a real data, or it was virtually generated. If the authors used real data, it should be explained in more details from which patients this data was adopted.

5.     Authors mentioned that blood was treated as Newtonian liquid. This assumption should be explained.

6.     The authors should stress in the discussion or conclusion part how their work can improve patient’s diagnostic. Also, the practical aspect of this study should be included.

Round 2

Reviewer 1 Report

The authors have considered the observations made and the article in its current form can be published in the journal

Reviewer 3 Report

The authors addressed all my to all my questions and comments. The article may be publised in its current form.